mental health; caregiver; carers; group interventions; mixed methods; implementation research; India

**Corresponding author:**
Kaaren Mathias;
Email: kaaren.mathias@canterbury.ac.nz

†Disha Agarwal and Christopher Bailie are equal first authors and contributed equally in writing this manuscript.

# Scaling a group intervention to promote caregiver mental health in Uttarakhand, India: A mixed-methods implementation study

Disha Agarwal[1†], Christopher R. Bailie[2†], Samson Rana[1], Laxman Balan[1], Nathan J. Grills[2] and Kaaren Mathias[1,3]

[1]Project Burans, Herbertpur Christian Hospital, Atten Bagh, India; [2]Nossal Institute for Global Health, University of Melbourne, Melbourne, VIC, Australia and [3]Te Kaupeka Oranga, University of Canterbury, Christchurch, New Zealand

## Abstract

Caregivers are integral to health and social care systems in South Asian countries yet are themselves at higher risk of mental illness. Interventions to support caregiver mental health developed in high-income contexts may be contextually inappropriate in the Global South. In this mixed-methods study, we evaluated the implementation and scaling of a locally developed mental health group intervention for caregivers and others in Uttarakhand, India. We describe factors influencing implementation using the updated Consolidated Framework for Implementation Research, and selected implementation outcomes. Key influencing factors we found in common with other programs included: an intervention that was relevant and adaptable; family support and stigma operating in the outer setting; training and support for lay health worker providers, shared goals, and relationships with the community and the process of engaging with organisational leaders and service users within the inner setting. We identified further factors including the group delivery format, competing responsibilities for caregivers and opportunities associated with the partnership delivery model as influencing outcomes. Implementation successfully reached target communities however attrition of 20% of participants highlights the potential for improving outcomes by harnessing enablers and addressing barriers. Findings will inform others implementing group mental health and caregiver interventions in South Asia.

## Impact statement

Nearly every disabled person in South Asia (such as people with difficulties walking or with mental health problems) is supported by family caregivers for their activities of daily living. Caregivers are central in the disability ecosystem in settings like India, where there is limited public and social support for disabled people. Yet caregiving is heavy and unceasing work and most caregivers receive little support meaning that they are at higher risk of experiencing mental distress. Few studies describe what works to strengthen the mental health and well-being of caregivers, and even fewer describe how an intervention can work well when rolled out among other organisations. This is essential to know in order to deliver interventions to larger numbers of people. This study tries to address this gap. In 2021, we had evaluated this caregiver mental health intervention (Nae Umeed or New Hope) and found it was effective. In this project, we aimed to describe the process of rolling out Nae Umeed with seven other organisations to learn what works well and what does not. We found Nae Umeed was a relevant and adaptable intervention that participants were positive about. Participation went better when participants also experienced family support to join in group sessions and try out the new ideas at home. We identified factors that made it work well from the perspective of organisations such as finding that the community health workers who facilitated groups needed to connect relationally with the community and to be supported with resources and training. Nae Umeed did not go as well if participants were too overloaded with other responsibilities, or if they experienced stigma. The group format allowed people to form new friendships which also provided social support after the programme finished. Nae Umeed merits consideration and rollout in other settings in South Asia.

## Introduction

Approximately one in six adults is caregivers (Tur-Sinai et al., 2020). Caregiving, defined as the provision of care to another person with a long-term care need outside of any formal framework (Tur-Sinai et al., 2020), is crucial to the sustainability of health and social care systems, but also can be associated with negative mental health consequences for the caregiver (Pinquart and

Sörensen, 2003; Talley and Crews, 2007). Caregivers in low- and middle-income countries (LMICs) have an estimated 50% higher odds of depression than non-caregivers, even after controlling for socioeconomic factors that may in part mediate mental health effects (Magaña et al., 2020).

In India, caregivers who are socioeconomically disadvantaged are additionally vulnerable to mental ill-health. Women report high caregiver "burden" and make up the largest proportion of caregivers (Kumar and Gupta, 2014; Mandowara et al., 2020; Madavanakadu et al., 2021). Additionally, caregivers who are socially isolated (Jagannathan et al., 2014; Mathias et al., 2019; Bapat and Shankar, 2021), under financial strain (Bapat and Shankar, 2021; Madavanakadu et al., 2021), or less educated (Jagannathan et al., 2014; Mandowara et al., 2020; Bapat and Shankar, 2021; Menon et al., 2022) are more vulnerable to adverse mental health. A further challenge is that access to quality affordable primary health care in India is limited, especially rurally (Patel et al., 2015). The importance of caregiver mental health in India is likely to rise as the population ages, the burden of non-communicable diseases increases, and there is limited responsiveness to emerging needs in the social welfare and health systems (Bollyky et al., 2017).

Relatively little progress has been made on strategies to support caregiver well-being and mental health in India. Globally, mostly in high- and middle-income settings, a variety of interventions to improve caregiver mental health have been developed (Sörensen et al., 2002; Hinton et al., 2019), however, these may be of limited utility in lower resource settings. In India, a few interventions have been trialled in single health services or districts (Das et al., 2006; Dias et al., 2008; Kulhara et al., 2009; Chakraborty et al., 2014; Chatterjee et al., 2014; Lamech et al., 2020; Baruah et al., 2021; Singh et al., 2021; Sims et al., 2022; Stoner et al., 2022), but have not been implemented at scale. Scaling-up of interventions in LMICs has been identified as a gap in implementation research (Meffert et al., 2016; Alonge et al., 2019).

*Nae Umeed* (New Hope) is a community-based group intervention aiming to improve caregiver mental health by building knowledge and skills in psycho-social health and social participation. The intervention was developed by Burans (2022), a partnership initiative focussed on improving mental health in communities in Uttarakhand, India. *Nae Umeed* was piloted in 2019 and 2020 in Dehradun, the capital of Uttarakhand (Bailie et al., 2023). Following this, in 2022, Burans scaled up implementation with other organisations in the Uttarakhand Community Health Cluster (CHGN-UKC). This study aimed to evaluate this scale-up implementation, with sub-objectives of describing the implementation and its outcomes and exploring barriers and enablers to implementation.

## Methods

### Intervention

*Nae Umeed* consists of 14 modules designed to be delivered within community settings to groups of 8–12 participants by trusted local community health workers. The curriculum covers self-care, caregiving, psychosocial well-being, behaviour management, drugs, accessing support and entitlements, and managing household finances. The intervention design is described in greater detail elsewhere (Emmanuel Hospital Association, 2019; Bailie et al., 2023). In a pre-post effectiveness study as part of the 2020 Dehradun implementation, participants showed significant improvement in self-rated depression, well-being, and social participation (Bailie et al., 2023), consistent with previous findings for the effectiveness of psychosocial group interventions (Sörensen et al., 2002).

### Implementation context and strategy

Implementation was conducted in Uttarakhand, North India, a state with a population of about 10 million people, of whom 70% live rurally, 20% are scheduled caste or tribe and 80% are literate. There is very limited access to outpatient care or counselling for people with mental health problems (Mathias et al., 2015). Women are typically responsible for most domestic and household tasks.

Burans implemented *Nae Umeed* in conjunction with CHGN-UKC, a partnership of community health and development non-governmental organisations (NGOs) that collaborate on training, resource development, advocacy, and research partnerships (Grills et al., 2016; Uttarakhand Community Health Cluster, 2022). The implementation strategy aimed to engage and support CHGN-UKC partners to deliver *Nae Umeed* within their existing work areas. CHGN-UKC partners were invited to participate in the implementation of Nae Umeed and seven partners participated.

Burans provided centralised technical assistance and training on the intervention and partners allocated community health workers (facilitators) to deliver the intervention. Facilitators were responsible for forming groups in their local catchments through existing community networks, conducting meetings, and recording attendance. Burans project staff supervised training, conducted support visits, and assisted with field issues. The training involved several one-day workshops spaced over the intervention. Facilitators were able to access support via a Whatsapp group. Burans paid an allowance to facilitators to cover travel costs.

Facilitators formed 16 groups with 158 participants across the districts of Dehradun (10 groups), Tehri (4 groups), and Uttarkashi (2 groups). Modules were delivered from April to September 2022. Groups in Tehri and Uttarkashi were formed in rural mountainous regions, where the main source of income is agriculture, and where there is limited access to quality education and healthcare. Groups in Dehradun were formed in both rural and semi-urban areas. The main source of income in these areas is daily wage labour and access to education and health care facilities is easier.

### Positionality

D.A., S.R. and L.B. are Indian nationals who speak Hindi as their first language and are employed by Burans, Herbertpur Christian Hospital (HCH). K.M. was employed by Burans, HCH for 11 years and speaks Hindi. K.M. and N.J.G. have both worked in Uttarakhand for more than a decade and are currently based in universities in Australasia. C.R.B. is an Australian national who worked as a public health medicine trainee with Burans for a six month period.

### Study design

Our observational mixed-methods evaluation consisted of (1) Qualitative exploration of implementation determinants using the updated Consolidated Framework for Implementation Research (CFIR) (Damschroder et al., 2022), and; (2) Quantitative assessment of implementation outcomes (reach, dose delivered) using administrative and survey data.

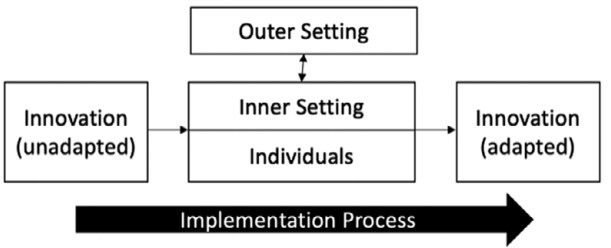

**Figure 1.** Schematic diagram of the Consolidated Framework for Implementation Research (CFIR). Adapted from Damschroder et al. (2009, 2022).

The CFIR is an implementation determinants framework consisting of 48 constructs across five domains covering the *Intervention* (*Nae Umeed*), and implementation context including *Outer Setting*, *Inner Setting*, *Individuals*, and *Implementation Process* (Damschroder et al., 2009, 2022; Figure 1). In operationalising the CFIR, we considered the *Inner Setting* domain to consist of Burans and implementing partners, and the *Outer Setting* domain to consist of the contexts within which they operate. Based on an initial scoping discussion with implementers we developed semi-structured in-depth interview (IDI) and focus group discussion (FGD) guides using the CFIR interview guide tool (CFIR Research Team, 2022) and a framework for the design and delivery of group interventions (Hoddinott et al., 2010) to include specific questions addressing the constructs: *Intervention design*, *Local conditions/attitudes*, *Available resources*, *Access to knowledge and information*, *Assessing needs and context*, and *Engaging*. We included open-ended questions to capture data on other constructs.

We defined reach (Glasgow et al., 2019) according to representation among participants of the disadvantaged demographic groups which are the primary targets of *Nae Umeed*. We operationalised the dose delivered (Rowbotham et al., 2019) per session, as the number of participants attending divided by the number of participants recruited at baseline. In reporting this study we adhered to the Standards for Reporting Implementation Studies (StaRI) Statement (Pinnock et al., 2017).

## Data collection

### Qualitative
D.A. and C.R.B. conducted IDIs (25 total) and FGDs (six total) with implementers (six IDIs), group facilitators (five IDIs, one FGD), and participants (14 IDIs, 5 FGDs) who were selected to reflect a range of stakeholder roles and intersectional identities. IDIs and FGDs lasted approximately 1 hour and were conducted in October and November 2022. We conducted FGDs and IDIs in person in Hindi, except for four IDIs with implementers which we conducted in English via videoconference. All IDIs and FGDs were audio recorded, transcribed, and translated into English where applicable. Transcripts were checked for accuracy by the interviewer.

### Quantitative
With participant consent, facilitators recorded anonymised demographic data from participants attending final meetings, using a standardised data collection instrument. We requested routine group attendance records from facilitators.

## Data analysis

### Qualitative
D.A. and C.R.B. assigned codes defined by the CFIR constructs and subconstructs, using Open Code 4.03 (ICT Services and System Development and Department of Epidemiology and Global Health, 2015). First, we each coded the same three transcripts, then met to review and resolve discrepancies. We coded the remainder of transcripts independently, with D.A., C.R.B., and K.M. meeting regularly to discuss coding decisions. We grouped coded data into categories defined by the CFIR domains. We interpreted data within each category by inductively developing themes relating to one or more implementation determinants.

### Quantitative
D.A. and C.R.B. analysed quantitative data such as attendance simply and descriptively.

## Ethics statement

Approval for this project was provided by the institutional ethics committee of the Emmanuel Hospital Association (protocol number: 240). Participants provided verbal and written consent to participate in data collection.

## Results

### Implementation determinants

Results of the qualitative analysis are presented under the domains of the CFIR framework. Within each domain, key themes are presented using a heading followed by a verbatim quote-linked running header. Table 1 summarises key themes and the identified implementation determinants within each domain.

### Intervention

Intervention relevance increased participation – "[*Nae Umeed*] *is related to our lives*".

Participants expressed favourable views on the *Intervention Design* (content and presentation of the intervention) (Damschroder et al., 2022) and found *Nae Umeed* content relevant and accessible which increased their enthusiasm for participation. One facilitator summarised this:

> [Nae Umeed] is related to our lives, women's lives. (It included) the things that we have to do every day like managing a budget and taking care of everyone's medications (Facilitator).

Participants described the most useful modules as those on self-care, medication, and financial inclusion. One woman described how she became more attentive with medication:

> After learning from here that we need to take medicines on time, I take my medicines on time and also give them to my child as well on time. Earlier, I was very careless with taking the medicines. Now, I am very meticulous (Participant).

Less favourable views of the intervention were expressed by some participants and facilitators in the context of *Intervention Relative Advantage* (comparison to other interventions in current use) (Damschroder et al., 2022). Many participants' families were involved in a Disability Inclusive Livelihoods Initiatives Project (DILIP), which offered economic opportunities for the families of people with disability by providing training in income-

**Table 1.** Summary of themes and implementation determinants identified through interviews and focus groups

| CFIR domain | Theme | Illustrative quote | Factors influencing implementation |
|---|---|---|---|
| Intervention | Intervention relevance increased participation | "NU is related to our lives" | • Local development and testing, perception of collaborative design<br>• Perception of content as practical and relevant<br>• Comparison to another project that offered financial opportunities |
| | Building and maintaining engagement through groups | "There was hesitation in the beginning, but now it's fine" | • Group dynamics: limited interaction in first few meetings, then cohesion as group relationships matured<br>• Informal recruitment via existing group members |
| Outer setting | Local contexts influenced participation | "Taking time out for them is challenging" | • Discretionary time for women linked with agricultural, household and caregiving responsibilities, especially in rural areas<br>• Travel time for participants to attend meetings in mountainous regions<br>• Availability of appropriate locations for groups activities<br>• Family and community attitudes towards the intervention<br>• Community stigma around disability and mental health<br>• Vulnerability of implementation model to weather events |
| Inner setting | Harnessing a network with common goals | "These are different partners coming together" | • Complex governance/accountability<br>• Competing priorities for facilitators<br>• Common focus on disability within partnership<br>• Established disability coordinator roles within partner organisations |
| | Providing centralised training and support | "One of the main gaps was in terms of training for the facilitators" | • Provision of centralised training over a large geographical area<br>• Degree of peer-support networks for facilitators<br>• Continuity in staffing for coordination<br>• Ongoing support for facilitators from the lead organisation |
| Individuals | Gender relations shaped group interactions | "If a female worker is there, they can share their problems" | • Agreement between participant and facilitator gender |
| Implementation process | Assessing partners' preparedness | "We could have spent more time in understanding where they are coming from" | • Assessing contexts and engaging with multiple partner organisations |
| | Local community engagement aided recruitment | "Some partners did very well, who had good rapport" | • Strength of facilitator-community relationships |
| | Adapting delivery to community needs | "We should work according to them" | • Adequacy of integrated referral mechanisms<br>• Flexibility in scheduling and delivery<br>• Adapting modules based on needs |
| | Monitoring intervention outcomes | "Now we have to see what changes" | • Feedback of success stories to partner organisation leads |

generating skills (e.g. agriculture) and small start-up donations (e.g. goats). Despite *Nae Umeed* and DILIP having distinct objectives, some facilitators suggested that DILIP shaped expectations of *Nae Umeed*, acting as a disincentive to participation. However, these suggestions in some instances more broadly reflected a general desire for more economic opportunities. One facilitator suggested:

> Since some members [of Nae Umeed] were already a part of DILIP, they were expecting some monetary help or donation. This impacted their will to join the groups (Facilitator).

Building and maintaining engagement through groups – "There was hesitation in the beginning, but now it's fine".

Participants described how groups provided safe environments for shared learning and cohesion. Group participants described sharing their problems and receiving support or advice from others who were going through similar challenges to themselves:

> People come here and share about their life. We listen to each other and give suggestions about what can be done. It feels good to listen to solutions. Everyone offers different suggestions (Participant).

In some cases, facilitators described poor engagement initially, which slowly improved as participants developed closer relationships and became more comfortable with the group:

> When the group started, people did not know each other so it was difficult, and they could not share much in the beginning. But towards the end, they built good relations with the participants, they could share what they felt with the group and that was helpful in bringing out new ideas (Facilitator).

Some participants subsequently brought friends or relations into the groups, resulting in informal "snowball" recruitment.

## Outer setting

Local contexts influenced participation – "Taking time out for them is challenging".

Participation was shaped by *Local Attitudes* and *Local Conditions* (beliefs, values, and socioeconomic/environmental conditions affecting implementation or delivery) (Damschroder et al., 2022). In rural areas, scheduling meetings was challenging due to prevailing gender relations which limit discretionary time for women with competing agricultural, household and caregiving responsibilities:

> […] the women are doing all the household things, looking after the children and husband's health and everything. Taking time out for them is challenging (Implementer).

Travel-associated time and financial burden were additional barriers for participants in the more remote mountain areas, where groups consisted of people from multiple villages.

> We have to walk 4–5 kilometres to come for the groups. Sometimes it becomes difficult as we have work in the fields. We cannot afford to take a jeep [public conveyance] (Participant).

Other practical constraints included finding a private space to meet:

> It was very difficult for us, as there is not that much space in anyone's house […] And when we used to do it in the outside area, people used to gather and ask questions (Participant).

Participants and facilitators described community conceptions of disability causation as either karmic or a "curse". Episodes of verbal harassment of people with disability were cited as evidence of community stigma. These attitudes resulted in some families being reluctant to engage in Nae Umeed.

> Earlier we were very apprehensive about sharing our daughter's condition with others and that is why I did not join the group. But now after listening to other women in the group I feel I am not alone. I feel confident and supported (Participant).

Implementers noted vulnerability of the implementation, monitoring and reporting to environmental *Critical Incidents* (disrupting large-scale events) (Damschroder et al., 2022) in the form of a heavy monsoon season.

> During the rainy season, due to bad weather conditions and landslides, it became difficult to make frequent visits to different villages for reporting and monitoring (Implementer).

## Inner setting

Harnessing a network with common goals – "These are different partners coming together".

Implementation was facilitated by *Mission Alignment* (alignment of implementation and delivery with overarching organisational goals) (Damschroder et al., 2022) with CHGN-UKC's focus on supporting people with disability. There were well-established *Relational Connections* (networks within and across *Inner Setting* boundaries) (Damschroder et al., 2022) in the form of groups within the cluster working on disability and mental health. An implementer described how organisational structures, such as having a disability coordinator within each partner, provided a workforce that already networked with families of people with disability:

> We already have disability coordinators in each project […] and they are normally involved in the disability program – so combining disability and mental health; that was a good idea (Implementer).

Quality *Communications* (information sharing practices) (Damschroder et al., 2022) within the cluster were used to spread awareness of the intervention and promote buy-in from partners. An implementer described how a regular cluster-wide meeting, "Linking to Learn" (L2L) allowed sharing of the intervention with a larger audience:

> […] the main purpose [of L2L] is that many organisations work in a remote area and organising, training, and updating is difficult for them. So CHGN can organise teaching and awareness training on health issues (Implementer).

The partnership model also provided challenges. Implementers described varying *Relative Priority* (importance of implementation compared to other initiatives) (Damschroder et al., 2022) among partner organisations. Given the coordinating organisation did not provide direct funding to partners, Burans had little influence over commitments to delivery. Facilitators had to balance delivery of Nae Umeed with routine work of their home organisations:

> As I am working with two different organisations, I have two different tasks to deal with, which are both important to me (Facilitator).

*Structural Characteristics* (infrastructure) (Damschroder et al., 2022) within Burans, including staffing changes, initially limited support visits, impacting partner engagement. This was addressed by engaging a designated coordinator and increasing the frequency of visits.

> Initially we [at Burans] were not able to hand over the implementation responsibility to a single person since the team was undergoing changes. But once a person was appointed, it all went smoothly (Implementer).

Implementers and facilitators reflected on tension between these characteristics of the *Inner Setting* and a perceived need to continue to support caregivers after completion of the intervention to sustain long-term outcomes:

> It is also a responsibility for Burans to ensure that people who have been connected through these groups remain connected. That is very important (Implementer).

To address this issue of sustainability, some suggested training group members to continue supporting caregivers after completion of the curriculum to maintain impacts of *Nae Umeed* (for example by forming self-help groups for micro-credit and savings):

> It is important for us to ensure that the group members hold meetings to maintain social support (Facilitator).

Providing centralised training and support – "A more systematic approach to training the facilitators would have worked better".

*Access to Knowledge and Information* (accessibility of guidance or training to implement and deliver the intervention) (Damschroder et al., 2022) was recognised as critical to implementation. Implementers and facilitators described the training as covering the intervention content but noted potential areas for training improvement. These included more focus on group formation and facilitation skills, and communicating the goals and purpose of the intervention. Logistic issues emerged with the initial training plan as described by one implementer:

> […] the facilitators from the hills come from very long distances. So coming for a one-day training, it meant like three days of their work. Which was not easy for them, to also go back and do the work they have in their organisation (Implementer).

In response to feedback, the implementation model was adapted so that some training was provided during monitoring visits. Ongoing support visits were valued and implementers viewed their role as supporting collaborative problem-solving with facilitators, rather than as monitoring fidelity or adherence to the intervention:

> [facilitators] should feel that they are supported by Burans. It should not seem to them that we are people who came from outside and just checked what was happening (Implementer).

### Individuals

Gender relations shaped group interactions – "It would be helpful to have a female facilitator as well".

Differences between the genders of facilitators (mostly male) and participants (mostly female), were described as limiting engagement between facilitators and participants on difficult topics or personal problems. While participants downplayed the importance of facilitator gender, implementers and facilitators offered varying viewpoints. Some described a reluctance of female participants to engage with male facilitators:

> Because if I'm a male person, [participants in villages] will not share feelings or the problems they're going through. If a female worker is there, they can share their problems and we can build good rapport with them (Implementer).

Others suggested that the ability of facilitators to build and maintain rapport, irrespective of gender, was more important in determining engagement.

### Implementation process

Assessing partners' preparedness – "We could have spent more time in understanding where they are coming from".

Implementers emphasised the importance of *Assessing Needs* and *Assessing Context* (collecting information about group priorities and barriers to implementation and delivery) (Damschroder et al., 2022) of the target population and partners to successful implementation. They described that further assessment with partners in more rural and remote districts might have led to modification of the implementation plan. Implementers suggested pre-implementation visits with partners and questionnaires for potential participants as possible strategies for *Assessing Needs*. For example, to accommodate travel difficulties, and varying facilitator backgrounds:

> I think if we could have spent more time in also understanding where these partners are coming from, and to understand more about their working methods, how they work in their organisations, to understand more of their context […] what we shared in orientation applied very well to some of the organisations, but with some I think it became more difficult (Implementer).

Implementers also reflected on the importance of *Engaging* leaders of partner organisations (encouraging participation in implementation) (Damschroder et al., 2022) to increase the visibility of work done by facilitators and ensuring that facilitators were valued within their organisations:

> […] if we plan more with leaders then we can work [more effectively] with disability coordinators from each organisation (Implementer).

Local community engagement aided recruitment – "Some partners did very well, especially those who had good rapport".

Participant *Engagement* was fostered by strong relationships between facilitators and community. Although some facilitators reported recruitment difficulties, many were able to recruit from their existing formal support networks and sometimes turned away potential participants because of high demand.

*Adapting* (modifying the intervention or setting for optimal fit) (Damschroder et al., 2022) was recognised as critical in maintaining participant engagement. As a result of barriers in the *Outer Setting*, including family and community attitudes and competing demands on time, participants often arrived late to meetings or sent another family member to attend. Facilitators described learning to incorporate flexibility in the delivery to maintain engagement.

> We taught them things, but we learnt from them as well. We understood that we cannot force them to come to the meetings at a particular time, but we need to take into consideration their suggestions as well. […] We understood that we should work according to them, only then will they understand us. […] There might be many other things that the person might be facing at the family level (Facilitator).

Facilitators described common modifications including adding content on health and medicines, and tips for saving money. A shortage of services drove some of these modifications. One facilitator reported feeling ill-equipped to deal with requests for help:

> We were not told this in the module that when people [ask questions about getting medication or a medical appointment] whom do we refer to and from whom we can expect help (Facilitator).

Facilitators often adapted the content from the manual using local language and dialects because some participants found the language complicated. They also made modifications such as the use of more interactive activities including games and visual aids.

Monitoring intervention outcomes – "Now we have to see what changes".

Notes recorded by facilitators on individual participant outcomes as part of *Reflecting and Evaluating* (collecting information about the success of implementation or intervention) (Damschroder et al., 2022) were reported to be useful in motivating partners.

> [The partners] all remain. And they have also seen, because whenever feedback from the women came, we all always gave feedback to the organisation head, and he/she was very pleased to see the changes (Implementer).

However, implementers emphasised the importance of planned monitoring following completion of the curriculum to assess sustained intervention and implementation impacts.

> […] after a few months when we'll revisit then we'll see […] we have to see what changes – we can see the changes in their attitude, or in their work, or in their connection with the person with disability (Implementer).

### Implementation outcomes

#### Reach

Sociodemographic data collection instruments were returned for 75 participants from 11 groups (Table 2), representing half the number of participants initially recruited. Three-quarters were female, half had completed less than 5 years of education, and one in five were of scheduled caste or tribe. Three-quarters reported having a person with disability in the household, most commonly a child of the participant.

**Table 2.** Sociodemographic characteristics of a sample of 75 *Nae Umeed* participants

| | *n* (% of total), unless otherwise specified |
|---|---|
| Age | |
| Median (range) | 38 (21, 75) |
| Missing[a] | 43 (57.3%) |
| Sex | |
| Female | 57 (76.0%) |
| Male | 18 (24.0%) |
| Caste | |
| General | 23 (30.7%) |
| Scheduled caste/tribe | 14 (18.7%) |
| Other backwards class | 38 (50.7%) |
| Religion | |
| Hindu | 54 (72.0%) |
| Muslim | 19 (25.3%) |
| Sikh | 2 (2.7%) |
| Years of education | |
| 0 | 16 (21.3%) |
| 1–4 | 21 (28.0%) |
| 5–9 | 29 (38.7%) |
| ≥10 | 9 (12.0%) |
| Location | |
| Rural | 35 (46.7%) |
| Rural – mountainous | 28 (37.3%) |
| Semi-urban | 12 (16.0%) |
| Household member with disability | |
| Participant | 21 (28.0%) |
| Child | 26 (34.7%) |
| Parent | 3 (4.0%) |
| Spouse | 3 (4.0%) |
| Sibling | 1 (1.3%) |
| Other relative | 2 (2.7%) |
| None | 19 (25.3%) |
| Type of disability in household | |
| Intellectual | 17 (22.7%) |
| Psychosocial | 13 (17.3%) |
| Difficulty walking or moving | 16 (21.3%) |
| Difficulty seeing or hearing | 10 (13.3%) |
| None | 19 (25.3%) |

[a]Age was not recorded for some participants due to an issue with collection forms.

### *Dose delivered*

Attendance was 75–80% of those who registered for the programme over the first nine modules, using data for a subset of 10 groups. Data were not available for the final five modules (Figure 2).

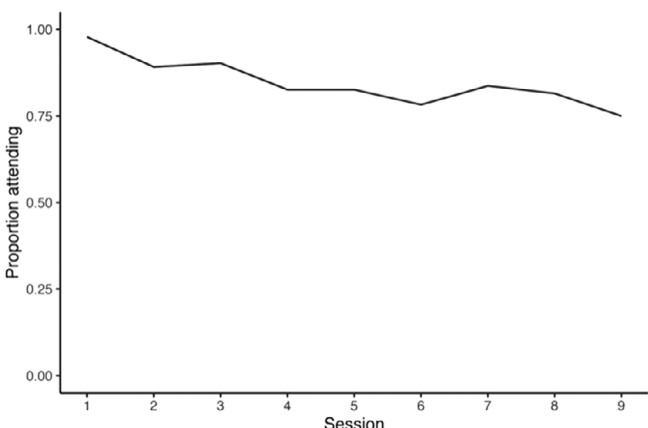

**Figure 2.** Overall attendance for the first nine *Nae Umeed* modules as a proportion of the number of participants recruited at baseline.

### Discussion

Several factors that influenced implementation in this study align with those described in the implementation of other mental health programmes in LMICs (Esponda et al., 2020; Greene et al., 2021; Le et al., 2022), including service user engagement, a relevant and adaptable intervention, family support, stigma, provider skills and connection with community, team relationships, common organisational goals, staffing, training, monitoring, and referral systems (Esponda et al., 2020, Greene et al., 2021, Le et al., 2022). Factors that we found were important which are less widely described in the literature included the influences of a group delivery format, competing responsibilities for caregivers in rural areas, challenges and opportunities associated with partnership delivery, and the varying influences of gender norms and relations across multiple CFIR domains. While the implementation successfully reached members of target vulnerable groups, issues in achieving sustained participant engagement highlight the potential for achieving better implementation and mental health outcomes by harnessing enablers and addressing barriers.

Reduction in participation to approximately 80% of baseline recruitment by the fourth module suggests that engagement was central for implementation success. Barriers to participation affecting some participants but not others, including lengthy travel, less supportive family, or expectations that were not met by the intervention may have led to selective early attrition. Fairly stable attendance following this early period (noting that attendance data was not available for the final five modules) may have been facilitated by the formation of supportive relationships within groups (Hoddinott et al., 2010), and adaptations to the implementation process to better support facilitators.

The group format of Nae Umeed may have facilitated improved mental health outcomes via multiple mechanisms. An increased sense of social connection and support was described by participants as resulting from the formation of new friendships. The group provided an opportunity to safely rehearse new skills (such as talk about emotions or speak of challenges), as well as a perception of collective strength from group membership. Our finding suggested these mechanisms acted to increase agency and action to access resources for mental health, consistent with other studies of psychosocial support group intervention in South Asian settings (Jordans et al., 2017; Morrison et al., 2019; Sikander et al., 2019). A group format may also increase the reach of an intervention

because fewer trained facilitators are required. Psychosocial support groups merit further attention in both research and the development of interventions as an effective, relevant, acceptable, and scalable platform to promote mental health in communities.

Our findings illustrate how effective delivery of group interventions requires careful consideration of facilitators and setting in the planning phase. Mental health interventions delivered by lay health workers can benefit from building on existing community relationships outside of formal healthcare settings (Kohrt et al., 2018), and available evidence suggests that they are effective in improving quality of life and day-to-day functioning (van Ginneken et al., 2021). However, as we found, they can place an additional burden on providers with other responsibilities (Kohrt et al., 2018), and rely on appropriate community settings for delivery which may not always be readily available (Puffer and Ayuku, 2022). Our findings also reinforce the need for ongoing support and training to maintain motivation and fidelity in the face of challenges related managing multiple responsibilities and engaging participants (for example discordance between facilitator and participant gender) (Wall et al., 2020).

Several key determinants identified in this study influenced participation through altering intervention demand and accessibility. Existing relationships between facilitators and communities aided recruitment. Intervention relevance, achieved through design in partnership with mental health workers and communities, facilitated retention. Determinants of service demand have been suggested as effective targets for implementing mental health interventions that are best addressed through community engagement to increase trust and achieve supply of interventions that are culturally relevant and sensitive to local needs (Greene et al., 2021). However, despite local design and piloting in Dehradun, scaling implementation to nearby areas brought a new set of barriers affecting demand including varying community and family attitudes, and travel and agricultural chores for participants. Although there is limited research in this area (Kohrt et al., 2018), our experience highlights the importance of an iterative and responsive approach to community engagement that continues through all phases of implementation.

Challenges in engaging caregivers of people with disability is a recurrent theme in the area of disability work in India, stemming from limitations to the time and resources of caregivers, and to reach of social and health services (Madavanakadu et al., 2021). Social isolation limits caregivers' awareness and utilisation of available services, and poor health among caregivers limits the care that a person with disability receives. Improving the well-being of vulnerable caregivers can help them to engage with service providers, facilitating the delivery of appropriate services to those with disability (Devassy et al., 2022). In this study, the implementation of Nae Umeed reached vulnerable target population groups including women and people with less education, although participants identifying as scheduled caste or tribe were similar to the Uttarakhand population. These findings reinforce the value of community-based models to engage with marginalised populations (Kohrt et al., 2018).

The partnership model in this study demonstrated how a cooperative approach across organisations could deliver a single intervention to support caregivers. NGOs can facilitate access to health services, particularly for the vulnerable, by providing free or low-cost community-based care (Sanadgol et al., 2021). Each partner in the delivery of Nae Umeed had existing disability interests, so were able to identify caregivers of people with disability in their community. However, NGOs typically work independently and

may have inadequate resources to undertake an intervention like Nae Umeed. In this example, the CHGN-UKC network worked collaboratively on implementation, a strategy that is important in India where there are more than 20,000 active non-profit health institutions (Central Statistics Office, 2012). The clustering approach of CHGN-UKC may help to increase access to resources, opportunities for collaboration, and credibility with the government (Grills et al., 2012; Safe et al., 2014; Grills et al., 2016). The CHGN-UKC cluster also provides an avenue for the findings presented here to be disseminated across providers and contexts.

The study also raises issues of the complexity in jointly delivering an intervention with different organisational entities, with little funding. Some partners were slow to implement due to competing priorities and lack of ownership of the intervention. The need for substantive engagement with partners and their leaders to support the implementation was evident in the findings of this study (Grills et al., 2012).

Our study had several limitations. The CFIR has been widely applied, in high-income settings (Damschroder et al., 2009; Kirk et al., 2016), yet may be less appropriate to LMICs where patterns or presentations of implementation determinants differ (Means et al., 2020). We felt that the updated CFIR (Damschroder et al., 2022) proved adequate to cover and classify determinants discussed in IDIs and FGDs. However, we recognise that our results primarily reflect determinants that we asked directly about or that study participants considered important. We found constructs related to individual characteristics less relevant, consistent with the experience of other authors, possibly due to organisational cultures having less focus on individuality in some LMICs (Means et al., 2020). Assessment of implementation outcomes was hampered by missing data for some groups, and attendance data for the final five modules. There is no reliable data on the demographics of caregivers in Uttarakhand, making our assessment of reach rudimentary. Assessment of other implementation outcomes, such as cost and fidelity, would help to inform decisions around the use of similar implementation strategies but were infeasible to assess in this evaluation.

This study contributes to evidence gaps in the implementation of mental health programmes in LMICs (Esponda et al., 2020). It is strengthened by the participation of service users in the evaluation and by implementation by a community-based organisation. It is relevant to the implementation of interventions that target caregivers, group interventions or those that focus mainly on prevention. The findings of this study can also inform others looking to implement group mental health and caregiver interventions in LMIC settings. There are few programmes and interventions targeting the needs of carers, although they are central to supporting people with psychosocial and other disabilities in LMIC (Hinton et al., 2019). Policymakers and national mental health and disability programmes should develop and implement interventions that increase mental health for carers (Hanlon et al., 2018), with a focus on studies of implementation processes. There are critical gaps in knowledge around what strengthens the psychosocial well-being of caregivers, particularly in LMIC, and implementation research should focus on interventions developed for local contexts (Bogart and Uyeda, 2009).

## Conclusions

In evaluating the implementation of a locally developed group caregiver mental health intervention in urban and rural North

India, this study identifies contextually relevant priorities for intervention and implementation strategy design. Supporting caregivers and their mental health yields positive outcomes for both carers and the family members with disabilities that they support. This is a high-priority group and improving their mental health and social participation requires active attention and coordinated action to support the psychosocial and socioeconomic needs of caregivers across the government and not-for-profit sectors.

**Open peer review.** To view the open peer review materials for this article, please visit http://doi.org/10.1017/gmh.2023.79.

**Data availability statement.** The data that support the findings of this study are available from the corresponding author upon reasonable request. The complete data are not publicly available due to their containing information that could compromise the privacy of research participants.

**Acknowledgements.** Thanks to the whole Burans team in Dehradun district for the support in implementing *Nae Umeed* and to the CHGN-Cluster for support in implementation at the peri-urban site. Thanks to the Institutional Ethics Committee of EHA for ongoing input to study design and to Herbertpur Christian Hospital for their ongoing support to all implementation and evaluation work.

**Author contribution.** Conceptualization: K.M.; Data collection: S.R., L.B.; Formal analysis: C.R.B.; Investigation: D.A., S.R., L.B.; Methodology: C.R.B., D.A., K.M.; Supervision: K.M.; Writing – original draft: C.R.B., D.A.; Writing – review and editing: N.J.G. and K.M. Note that D.A. and C.R.B. are joint first authors.

**Financial support.** Research costs were covered by existing program funds.

**Competing interest.** The authors declare that they have no competing interests in the publication of this article.

**Ethics statement.** Approval for this project was provided by the institutional ethics committee of the Emmanuel Hospital Association (protocol number: 240).

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
