## [Reviewer Report]

COVER LETTER

Dear Editors of Global mental health,

We have chosen your journal to submit this paper as it highlights the contribution and value of implementation research to understand interventions, and is a noted gap in low and middle income countries. It is also well-paired with our pre-post evaluation of the same intervention which is currently under 2nd review with your journal. 

First, this paper identifies the importance of context for intervention and implementation strategy design. It is based on a locally developed mental health intervention for caregivers in North India. It will inform others looking to implement group mental health and caregiver interventions in South Asian settings. 

Secondly, this study focuses on implementation barriers and enablers – using the updated Consolidated Framework for Implementation Research (CFIR) which increases the scalability, generalisability and relevance of the findings and informs others looking to implement group mental health and caregiver interventions in LMIC and South Asian settings. We additionally focused on domains like group delivery format, competing responsibilities for caregivers and challenges and opportunities associated with the partnership delivery model.

Thirdly, the study has been undertaken by a multidisciplinary, multi-country authorship team. DA, LB and SR are social work based mental health practitioners based in a non-profit organisation in North India; KM works and researches in community mental health in India and New Zealand, NG and CB are public health physicians working between Australia and India. 

Please note that the position of first authorship is with equal contribution by Disha Agarwal and Chris Bailie and so we request you note that these are joint first authors.

Thank you for your consideration of this paper for publication. 

Kaaren Mathias (corresponding and senior author)

---

## [Reviewer Report]

The authors have followed a good practice of being transparent about the findings from the study. They have also disclosed less favorable views expressed by some participants and facilitators in the context of Intervention Relative Advantage. It has identified the need to integrate mental health with existing disability related initiatives. The study highlights the need of group formation and facilitation skills, communicating the goals and purpose of the intervention for trainer and ongoing support visits rather than monitoring fidelity. Role played by gender, need for pre-implementation visits, incorporate flexibility in the delivery, managing queries outside the content etc. were the other interesting insights. The discussion is unique in highlighting need for developing iterative and responsive approach rather than replication as it is. The complexity in joint delivery combined with limitations on funding and varying priorities are explained well. The lack of cost and fidelity assessments may attract less interest from funders and other interested partners. Overall, this article serves well towards bridging the gap in literature, especially in LMIC context, about developing and studying the interventions improving caregivers' wellbeing.

---

## [Reviewer Report]

Well written article.

1. I would encourage the authors to do a spell check and grammar check (particularly the abstract).

2. It will help the readers to understand the findings better if the authors can consider a diagramatic representation of their CFIR framework findings

---

## [Reviewer Report]

Dear Editors of Global Mental Health,

Thank you for your consideration of this manuscript for publication. Please find our attached revised manuscript (clean) and response to reviewers.

Please note that we couldn’t edit the author list but one co-author has an updated email contact.

thank you again

Kaaren Mathias (on behalf of the author team).

Mr Samson Rana - samson.burans2023@gmail.com is the correct email ID.